# HYBRID ACNS: UNIFYING AUTO-COMPRESSING AND RESIDUAL ARCHITECTURES

## ABSTRACT

We propose **Hybrid Auto-Compressing Networks (H-ACNs)**, unifying ACNs and ResNets under a single mathematical formulation controlled by trainable scalar residual weighting parameters per layer. Through theoretical analysis, we show that both architectures represent points on a continuous spectrum, with traditional ACNs and ResNets as special cases. Our key contribution is demonstrating that H-ACNs, when initialized close to ACNs, match ResNets training efficiency while preserving ACN-like robustness and compression capabilities. Experiments across vision transformers, MLP-mixers, and GPT-2 architectures show that H-ACNs achieve training convergence on par with ResNets, while maintaining ACNs superior noise robustness and generalization. Furthermore, we discover that learned residual weights exhibit distinct connectivity patterns across tasks, namely, vision tasks favor local connectivity patterns resembling early visual cortex processing, while language tasks converge to modular hierarchical inter-layer structures similar to hierarchical language processing regions. We also examine how initialization impacts performance and connectivity, challenging the universality of the common ResNet-like initialization of residual weights. Overall, our results establish Hybrid ACNs as a practical framework for efficiently balancing training speed and representation quality, while revealing principles of how functional connectivity patterns should vary across domains, modalities, and tasks.

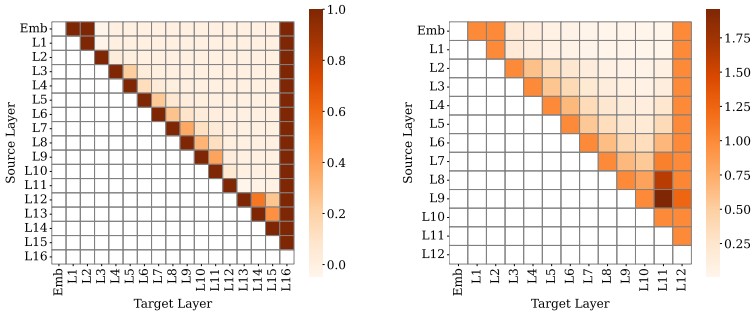

(a) **Learned Inter-Layer Connectivity in *Vision* Classification Models**

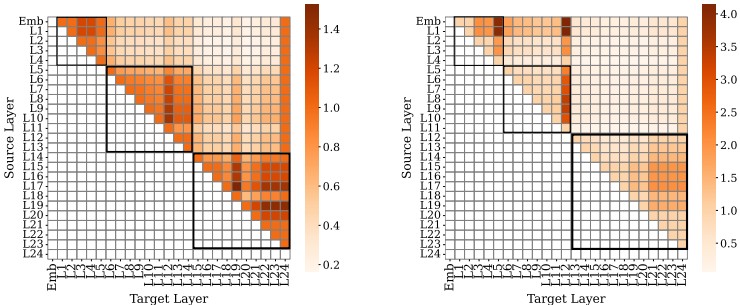

(b) **Learned Inter-Layer Connectivity in *Language* Models**

Figure 1: Entry *C[i][j]* (Sec. 2): direct connection strength from source layer $i$ to target layer $j$.

## 1 INTRODUCTION

Deep neural networks face a fundamental tradeoff between learning robust, generalizable representations and achieving efficient, stable training (Bengio et al., 1994; Balduzzi et al., 2017; Zhang et al., 2024). While architectures that enhance robustness (Yang et al., 2020) or promote rich feature learning, e.g., via dense connectivity, often produce superior representations and better generalization (Huang et al., 2017), they typically suffer from training instabilities, slower convergence, or computational overhead (Srivastava et al., 2015; Yang et al., 2020). Conversely, architectures optimized for training efficiency, such as residual networks, enable rapid convergence and stable gradient flow (He et al., 2016) but may underutilize network depth and produce less robust representations (Zhang et al., 2024; Lad et al., 2025; Csordás et al., 2025; Yang et al., 2020). This tradeoff between representation quality vs training efficiency remains a central challenge in neural architecture design.

Recently, Auto-Compressing Networks (Dorovatas et al., 2025) (ACNs) have been introduced, replacing residual connections with direct long feedforward connections from each layer to the output. Unlike traditional feedforward or residual architectures, ACNs enable automatic compression during training; networks naturally learn to concentrate critical information in early layers while deeper layers become redundant for simpler tasks, all without external pruning or regularization. Further, ACNs achieve better representational quality through enhanced noise robustness, superior generalization in low-data regimes, and improved continual learning capabilities . However, this architectural design comes at a significant cost: ACNs suffer from substantially slower training convergence and reduced training stability compared to their residual counterparts, making them less practical for large-scale applications despite their representational advantages.

This raises a fundamental research question: can we create a unified architecture that smoothly interpolates between ACNs and ResNets, capturing the compression capabilities and representational advantages of ACNs while maintaining the training stability and efficiency of residual networks? Furthermore, if we allow such a hybrid architecture to learn its own connectivity patterns through trainable interpolation parameters, what architectural structures emerge across different domains and tasks? Do vision and language tasks converge to fundamentally different connectivity patterns, and what do these learned structures reveal about the functional connectivity requirements of different cognitive tasks? And also, how does the initialization in the connectivity space—that is, the connectivity inductive biases—shape their behavior? These questions build on recent work exploring learnable dense connectivity, including depth-weighted averaging in transformers (Pagliardini et al., 2024), attention-based layer fusion methods (ElNokrashy et al., 2022) and neural architecture search approaches that learn optimal residual connection through trainable weights (Pham et al., 2018; Wang et al., 2023) and connects this direction with the auto-compression property (Dorovatas et al., 2025).

To address these questions we propose Hybrid Auto-Compressing Networks (H-ACNs), which unify ACNs and ResNets under a single mathematical formulation controlled by trainable scalar residual weighting parameters per layer. Through theoretical analysis, we demonstrate that both architectures represent points on a continuous spectrum, with traditional ACNs and ResNets as special cases. Our main contributions are:

- **H-ACNs achieve ResNet-like training efficiency while preserving ACN-like robustness, compression capabilities, and superior generalization** across vision transformers, MLP-mixers, and GPT-2 architectures. This translates into better downstream performance for complex tasks.

- **Learned residual weights exhibit distinct connectivity patterns across tasks and modalities**: vision tasks converge to local connectivity patterns resembling early visual cortex processing, while language tasks develop modular hierarchical inter-layer structures similar to hierarchical language processing regions as shown in Fig. 1.

- We find that initialization encodes powerful architectural priors that determines the final structure and behavior of the network, making the starting point as critical as the search algorithm itself. From this perspective, **one of our key contributions is the integration of learnable architectures with the auto-compressing inductive bias at initialization**.

## 2 HYBRID ACNs: INTERPOLATING BETWEEN ACNs AND RESNETS

ACNs and ResNets represent two instances of multi-path architectures, with distinct inter-layer connectivity patterns but a common output $y$ summation formula:

$$x_i^{ACN} = f_i(x_{i-1}^{ACN}), \qquad x_i^{Res} = f_i(x_{i-1}^{Res}) + x_{i-1}^{Res}, \qquad y = x_0 + \sum_{i=1}^{L} f_i(x_{i-1}) \qquad (1)$$

In ResNets, the final sum $y$ is implicit, arising from the residual accumulation at each layer, as each layer adds its output to the residual stream. A closer look at their equations indicates that they can be unified under a single mathematical formulation, controlled by a scalar residual weighting parameter for each layer, forming **Hybrid-ACNs** (H-ACNs):

$$x_i^{HACN} = f_i(x_{i-1}^{HACN}) + a_i x_{i-1}^{HACN}, \qquad y^{HACN} = x_0 + \sum_{i=1}^{L} f_i(x_{i-1}^{HACN}) \qquad (2)$$

Specifically, setting $a_i = 0$, $\forall i$, corresponds to a vanilla ACN, while $a_i = 1$, $\forall i$, recovers a vanilla ResNet [1]. Intermediate values of $a_i$ produce architectures that interpolate between the two, resulting in distinct connectivity patterns determined by the residual weights. By making these weights trainable, the network can dynamically learn its **inter-layer connectivity** during training.

Across architectures, the **input to layer k** as a function of the outputs of previous layers can be expressed as:

$$input_k = \sum_{i=0}^{k-1} c_{i \to k} \, h_i \qquad (3)$$

where $h_i = f_i(x_{i-1}), \forall i > 0$ is the output of layer $i$, $h_0$ denotes the input embedding and $c_{i \to k}$ denotes the **strength of the direct residual connection from layer** $i$ **(source) to layer** $j$ **(target)** (e.g., via shortcut or residual pathways). We can then calculate $c_{i \to j}$ for all layers and store them in the **Direct Layer Connectivity Matrix**, a square matrix $C \in \mathbb{R}^{L+1 \times L+1}$, where $L$ is the number of layers and $C[i][j] = c_{i \to j}$. Therefore, in this structure, **column** $k$ of $C$ corresponds to the vector $\mathbf{c^k} = [c_{0 \to k}, c_{1 \to k}, \dots, c_{(k-1) \to k}, 0, \dots, 0]$, which stores all weights of Eq. 6. By default, $c_{(k-1) \to k} = 1$ as successive layers are always connected by the direct feed-forward connections in all considered architectures. Under this definition, it holds that:

- in standard FFNs each layer $j$ receives input only from layer $j-1$, resulting in $C[j-1][j] = 1$ and all other entries zero (Fig. 8b),

- in standard residual architectures, $C[i][j] = 1, \forall i < j$ (Fig. 8a),

- for ACNs, we have $C[j-1][j] = 1$ and $C[i][L] = 1, \forall i < L$; the rest being zero (Fig. 8c).

For H-ACNs, in order to avoid $O(L^2)$ residual weight parameter growth and the $O(L)$ additional memory required during the forward pass (to store the outputs of all layers independently), we introduce $L$ learnable residual weights. These weights act synergistically, enabling direct inter-layer connections through multiplicative interactions:

$$c_{i \to j} = \prod_{l=i+1}^{j-1} a_l \qquad (4)$$

To further illustrate how this equation is derived, we revisit Eq. 2 and expand, as an example, the input of layer 4 as a function of the outputs of all preceding layers:

$$x_4 = h_3 + a_3 x_3 = h_3 + a_3(h_2 + a_2 x_2) = \dots = \underbrace{1}_{c_{3 \to 4}} h_3 + \underbrace{a_3}_{c_{2 \to 4}} h_2 + \underbrace{a_3 a_2}_{c_{1 \to 4}} h_1 + \underbrace{a_3 a_2 a_1}_{c_{0 \to 4}} h_0 \qquad (5)$$

where $x_i$ and $h_i$ are the input and output of layer $i$, respectively, and $h_0$ the initial input embedding.

---

[1] In Appendix G, we provide a pseudo-implementation of the H-ACN forward pass, unifying ResNets and ACNs forward passes, for additional clarity.

As discussed previously, ACNs and ResNets represent two extreme points of multi-path network architectures in terms of the paths available within the network. Following the analysis of (Veit et al., 2016) and as argued in the original ACN paper, ACNs have a number of paths that grows linearly with the number of layers, whereas residual networks exhibit an exponential growth in paths. H-ACNs interpolate between these two extremes and can behave more like residual or auto-compressing architectures depending on the structure of the matrix $C$ determined by the learned residual weights. These weights act as gates, modulating signal flow: for small to medium $a$ values, forward signals naturally attenuate, preserving the layer-wise characteristics of ACNs. For more complex tasks, the residual gates can open (more), allowing strong information flow to deeper layers and improving training and gradient propagation.

Importantly, this gate tuning and the resulting information flow dynamics are learned internally by the network during optimization, since the residual weights are trainable. Training such learnable architectures constitutes a dynamic system that converges to different behaviors depending on initialization. As we hypothesize and validate empirically, different initializations of the $\alpha$ parameters — i.e., imposing either an auto-compressing or residual inductive bias — lead to distinct dynamics, final behaviors, and connectivity patterns.

## 3 EXPERIMENTS

In this section, we implement and test the proposed H-ACN architecture in a variety of tasks, modalities and architectures, ranging from image classification (CIFAR-10 (Krizhevsky, 2009), ImageNet (Russakovsky et al., 2014)) to language modeling (OpenWebText2 (Gao et al., 2020), PG-19 (Rae et al., 2019)). We consider MLP-Mixer (Tolstikhin et al., 2021) and Transformer (Vaswani et al., 2017; Dosovitskiy et al., 2020) models and compare H-ACNs against vanilla Residual and vanilla ACN architectures [2]. This section is organized as follows: Subsection 3.2 covers MLP-Mixer on CIFAR-10, Subsection 3.3 discusses ViT on ImageNet, and Subsection 3.4 details GPT-2 pre-training.

### 3.1 EXPERIMENTAL SETUP

**Layer-wise accuracy.** This metric is our primary tool for evaluating the performance of intermediate layers and auto-compression following (Dorovatas et al., 2025). Layer-wise accuracy for layer $i$ refers to the accuracy obtained by performing a forward pass up to layer $i$, treating it as if it were the final layer and feeding it into a common head trained on the full network.

**Initialization of residual weights.** Across all experiments, we initialize the residual weights from the normal distribution $\mathcal{N}(0.25, 0.005)$. This initialization places the network close to a vanilla ACN, effectively imposing the auto-compressing inductive bias. In Appendix D we ablate other choices involving depth-wise initialization, while in Section 4 we further explore the behavior under different mean values. We note that, in this work, we focus on investigating the behavior and performance of the learnable architecture when it is initialized closer to an ACN vs a ResNet. More complex initialization schemes (e.g., layer-dependent) or alternative training strategies of the residual weighting are left for future work [3].

### 3.2 BRIDGING AUTO-COMPRESSION AND TASK LEARNING

We begin by integrating the three architectures into a 16-layer MLP-Mixer and training on CIFAR-10 classification dataset for 300 epochs. Additional training details and hyperparameter settings are provided in Appendix A. In Figure 2a, we show the Layer-wise Accuracy of the three variants. We observe that **H-ACNs achieve performance comparable to the residual architecture *at the same training steps*, while simultaneously exhibiting auto-compression behavior similar to ACNs**. Furthermore, in Figure 2b, which presents the training loss over epochs, we find that H-ACNs demonstrate a significant advantage over ACNs in training speed, outperforming even the residual architectures. Notably, ACNs require approximately 100 additional epochs to match the

---

[2] In Appendix B we further compare H-ACNs with other recent residual architectures.

[3] Another technique we employ to enhance the forward pass of H-ACNs and ACNs is *Depth-Adaptive LayerNorm*; further details and an ablation study of its effect are provided in Appendix D.

performance of H-ACNs and Residuals. Overall, H-ACNs effectively combine the strengths of both architectures, bridging the gap between auto-compression and efficient learning.

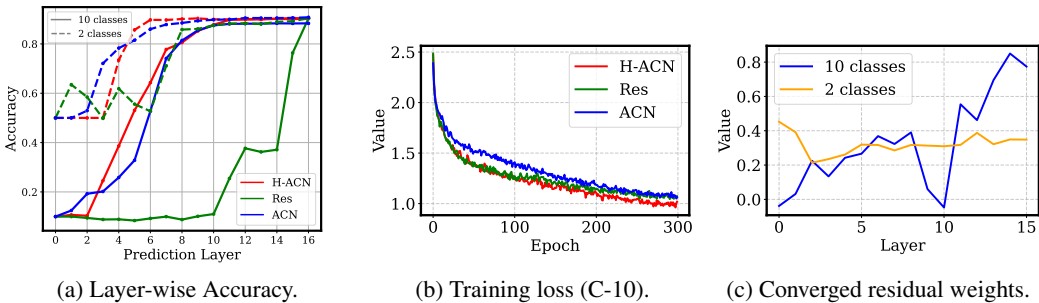

(a) Layer-wise Accuracy.      (b) Training loss (C-10).      (c) Converged residual weights.

Figure 2: **MLP-Mixer on CIFAR-10.**

### 3.2.1 CONVERGED CONNECTIVITY VS TASK DIFFICULTY

As shown in (Dorovatas et al., 2025), auto-compression is a function of task difficulty, measured there as the number of classes in a classification task. Here, we replicate this experiment to examine the behavior and converged connectivity of the H-ACN model as task difficulty varies. Intuitively, we expect that as task difficulty decreases, (1) the auto-compression of H-ACNs will increase, similar to the behavior observed in ACNs, and (2) the converged connectivity of H-ACNs will more closely resemble that of ACNs in the connectivity space, with $\alpha$ values (residual weights) being smaller, effectively revealing redundancy in the predefined architecture.

To validate this intuition, we modify the experiment by reducing the number of CIFAR-10 classes from 10 to 2 and training until the models achieve comparable performance to the 10-class case for a fair comparison. As shown in Figure 2a (dashed lines), we confirm that for the two-class setting, the auto-compression of H-ACNs increases mirroring ACNs behavior. Interestingly, H-ACNs seem to achieve stronger auto-compression in this case, further highlighting the advantages of learnable architectures. Furthermore, Figure 2c shows that the converged residual weights have smaller magnitudes in the binary classification case, indicating that this task needs less capacity compared to the 10-class case. Another perspective, which we will also analyze later, is that *as task difficulty increases, the intra-connectivity of the network also increases* [4]. In summary, we observe a smooth interplay between ACNs and residual networks, with task difficulty acting as a key factor influencing the balance between these architectures in the learned connectivity of H-ACNs.

### 3.3 HYBRID-AC VISION TRANSFORMERS ARE COMPACT AND ROBUST CLASSIFIERS

Next, we consider more challenging tasks and larger architectures, specifically the Vision Transformer (Dosovitskiy et al., 2020) on the Imagenet-1k (Russakovsky et al., 2014) classification dataset. We integrate the three architectures into a 12-layer ViT and train for 300 epochs, following the setup described in (Beyer et al., 2022). We report the final top-1 accuracy (Figure 3b) to evaluate the generalization capabilities of the architectures, and plot accuracy versus epochs (Figure 3a) to analyze their training dynamics.

From the accuracy vs epochs plot we observe that H-ACNs match the training speed of residual networks, being trained significantly faster than ACNs (that require approximately 700 epochs to match the performance of the counterparts). Interestingly, we find that H-ACNs achieve the best top-1 final accuracy, outperforming both ACNs and ResNets. Results highlight the effectiveness as well as efficiency of interpolating between the two vanilla architectures.

**Robustness against input noise:** One of the arguments for ACNs learning better and more robust representations is that they have been shown to be more resilient to input noise compared to residual networks. Here, we examine the behavior of H-ACNs under Gaussian input noise of zero mean and

---

[4]The magnitudes of the residual weights determine the inter-layer connectivity; for a more detailed analysis we refer to Sec. 4.

(a) Accuracy vs Epochs.

| Models | Acc. |
|--------|------|
| Res | 78.77 |
| ACN | 78.69 |
| H-ACN | **79.2** |

(b) Best final Accuracy.

| Noise | Res | H-ACN |
|-------|-----|-------|
| 0 | 78.77 | 79.2 |
| 0.1 | $76.57 \pm 0.03$ | $77.02 \pm 0.05$ |
| 0.3 | $66.89 \pm 0.03$ | $67.59 \pm 0.04$ |

(c) Performance under **input zero-mean Gaussian noise**.

Figure 3: **ViT on ImageNet.**

varying standard deviation. Across all noise levels, we find that H-ACNs consistently outperform residual networks, providing further evidence that H-ACNs successfully combine the advantages of both architectures. To summarize this ViT/Imagenet experiment:

**H-ACNs achieved training speeds comparable to residual networks while showing stronger generalization and improved noise robustness, akin to ACNs**.

### 3.4 CAUSAL LANGUAGE MODELING WITH HYBRID-AC DECODERS

Finally, we explore the effect of inter-layer connectivity patterns in auto-regressive language modeling. Specifically, we want to examine how each architecture (Res, ACN, H-ACN) affects the training dynamics and the downstream zero-shot performance of language decoders.

**Models.** We consider GPT-2 (Radford et al.) style decoder models with $L = 24$ layers ($\sim$210M params), RoPE (Su et al., 2024) positional embeddings, maximum sequence length of 256 and embedding dimension $d = 768$. All experimental details can be found in Appendix A.

**Datasets.** For auto-regressive pre-training, we primarily use the OpenWebText2 dataset (Gao et al., 2020) (OWT-2) which consists of around 17B tokens. We also pre-train on PG-19, consisting of full-length books published over 100 years ago and extracted from Project Gutenberg (Rae et al., 2019). We use both dataset to investigate the learned connectivity patterns of H-ACNs as a function of the nature of the pre-training data, contrasting the literary and dated content of PG-19 with the more factual, diverse and contemporary web-based content of OpenWebText2.

For zero-shot evaluation of the pre-trained models, we consider popular downstream benchmarks, namely HellaSwag (Zellers et al., 2019) (commonsense inference with grounded scenarios), PIQA (Bisk et al., 2020) (physical reasoning about everyday situations), and ARC-E (Clark et al., 2018)(grade-school level multiple-choice questions testing scientific and logical reasoning).

**Implementation Details.** We consider residual and Hybrid ACN architectures [5]. All models are trained for 240K steps with a batch size of 128, totaling approximately 30B tokens seen during training. We use the AdamW (Loshchilov & Hutter, 2017) optimizer with cosine learning rate scheduling and warmup; detailed training hyperparameters are provided in Appendix A.

**Results.** Figure 4a and 4b show the layer-wise validation perplexity [6] of Residual and Hybrid ACN models in log format [7]. We see that in both cases, H-ACN performance is on par with the ResNet but with significantly improved intermediate layer perplexity [8]. Then, we evaluate the pre-trained

---

[5]We found that for the same training steps ACNs significantly underperform ResNets and H-ACNs, and thus we do not include them in the results.

[6]Computed in the same way as layer-wise accuracy.

[7]We do this for visualization clarity; in the early and intermediate layers, Res PPL values are really large compared to H-ACN.

[8]Top-1 val perplexity for H-ACN vs Residual is 19.78 vs 19.82 on OWT-2 and 16.78 vs 16.65 on PG-19.

models' zero-shot downstream capabilities to assess how well they generalize. For this, we choose the pre-trained on OWT-2 dataset models, since it is modern and factual, thus closer to the nature of the downstream tasks [9]. In Table 4c, we observe that H-ACNs show improved average downstream performance of $41.8\%$ compared to $41.2\%$ of the vanilla residuals, further **highlighting the stronger generalization capabilities of H-ACNs**.

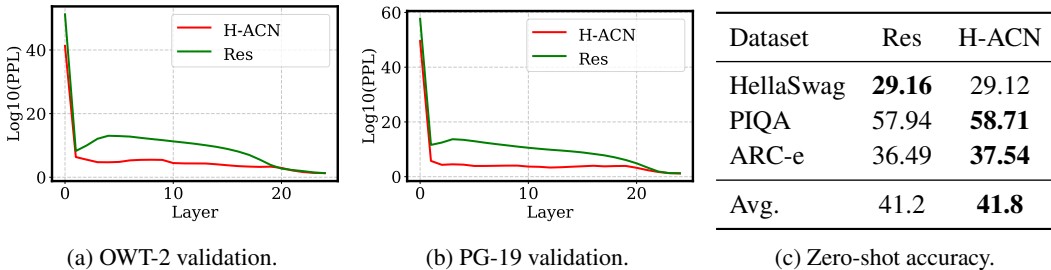

| Dataset | Res | H-ACN |
|---------|-----|-------|
| HellaSwag | **29.16** | 29.12 |
| PIQA | 57.94 | **58.71** |
| ARC-e | 36.49 | **37.54** |
| Avg. | 41.2 | **41.8** |

(a) OWT-2 validation.  (b) PG-19 validation.  (c) Zero-shot accuracy.

Figure 4: **GPT-2 models (L=24). (left & middle)** We plot the final validation perplexity in log format (for visualization purposes) of all intermediate layers of the two models for (left) OpenWebText-2 (OWT-2) dataset and (middle) PG-19 dataset. **(right)** We show the zero-shot performance of the two models on various downstream datasets, when pre-trained on OWT-2.

**Robustness against input character-level noise.** We test the robustness of the pre-trained models on OWT-2 against character-level noise on HellaSwag. For each character, with probability $p$, we either remove it, insert another character, or swap it with the subsequent character. We find (Fig. 1) that H-ACNs again outperform residual networks under noisy conditions, **further extending the previously observed noise-robustness characteristics of auto-compressing architectures to the language domain**.

| Noise ($p$) | Res | H-ACN |
|-------------|-----|-------|
| 0 (w/o noise) | 29.16 | 29.12 |
| 0.01 | $27.55 \pm 0.15$ | $27.95 \pm 0.20$ |
| 0.03 | $26.10 \pm 0.22$ | $27.00 \pm 0.18$ |

Table 1: **GPT-2 (L=24).** Performance on HellaSwag under different character-level (insert/delete/swap) noise levels.

## 4 THE EMERGENCE OF STRUCTURE: ANALYSIS OF THE RESIDUAL WEIGHTS

Having demonstrated that H-ACNs achieve training efficiency comparable to ResNets while learning superior representations with enhanced robustness and task-adaptive compression, we now examine the learned inter-layer connectivity structures that underlie these properties. This section analyzes how connectivity patterns are shaped by residual weight initialization, task complexity, data modality, and training dynamics.

To quantify these connectivity patterns, we define the **total connectivity strength** $\Gamma$ as $\|\boldsymbol{\alpha}\|_2/\sqrt{L}$, where $\boldsymbol{\alpha}$ represents the vector of learned residual weights and $L$ is the number of layers. This normalized magnitude serves as a scalar proxy for the overall strength of direct inter-layer connectivity taking values $\Gamma = 0$ for ACNs and $\Gamma = 1$ for ResNets. Additionally, we analyze the full connectivity structure through the Direct Layer Connectivity Matrix $\mathbf{C} \in \mathbb{R}^{(L+1) \times (L+1)}$, where $\mathbf{C}[i][j] \equiv c_{i \to j}$ represents the connection strength from source layer $i$ to target layer $j$.

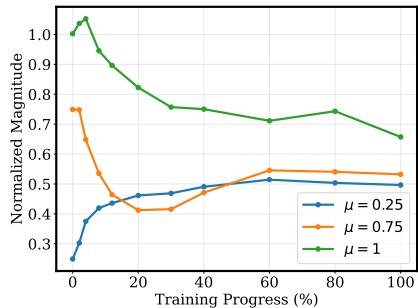

Figure 5: **MLP-Mixer/CIFAR-10.** Evolution of total connectivity strength $\Gamma$ under different inits $\alpha_i \sim \mathcal{N}(\mu, 0.005)$.

---

[9]We also tested PG-19 pre-trained models and we observed poor zero-shot performance.

**Residual weights initialization.** The main H-ACN parameter is the mean $\mu$ of the distribution that initializes the residual weights $\alpha_i \sim \mathcal{N}(\mu, 0.005)$, controlling the interpolation between ACN ($\mu = 0$) and ResNet ($\mu = 1$) regimes. Throughout our experiments, we initialize with $\mu = 0.25$, placing H-ACNs closer to the ACN regime. To understand how this choice affects performance, we test larger values of $\mu$ on the Mixer/CIFAR-10 setup. As shown in Table 2, performance consistently deteriorates for larger values of $\mu$. Examining the evolution of total connectivity strength $\Gamma$ in Fig. 5 shows that ResNet-like initializations lead to more densely interconnected networks throughout training. Importantly, starting from a sparser connectivity pattern appears to yield better performance, suggesting that the network favors evolving from sparse to dense connectivity during training.[10]

**Robustness of initialization.** To evaluate the robustness of our connectivity initialization, we compute the Pearson correlation of the converged residual weights across independent training runs. On GPT-2/OWT-2, we observe an average correlation of $95\%$, while on MLP-Mixer/CIFAR-10 we obtain $92\%$, suggesting that our initialization scheme consistently guides training toward stable connectivity solutions.

**Evolution of the connectivity during training.** We examine how residual weights evolve during training by tracking: (1) the **convergence distance** $\Delta_t \equiv \|\boldsymbol{\alpha}_T - \boldsymbol{\alpha}_t\|_2$, measuring how far current residual weights $\boldsymbol{\alpha}_t$ are from their final values $\boldsymbol{\alpha}_T$, and (2) total connectivity strength $\Gamma$ to capture connectivity dynamics. The evolution of $\Gamma$ (Fig. 6b) reveals rapid growth during early training, followed by stabilization or slight decline around the 60% mark. This pattern suggests that H-ACNs first undergo architectural exploration while learning the task, then shift to task-focused optimization once connectivity structure converges. The trajectory of $\Delta_t$ (Fig. 6a) reveals similar dynamics.[11]

| Init of $\alpha$ | Acc. |
|---|---|
| $\mu = 0.25$ | 90.2 |
| $\mu = 0.75$ | 89.7 |
| $\mu = 1$ | 88.7 |

Table 2: **MLP-Mixer/CIFAR-10**. Best final accuracy under different initializations $\alpha_i \sim \mathcal{N}(\mu, 0.005)$.

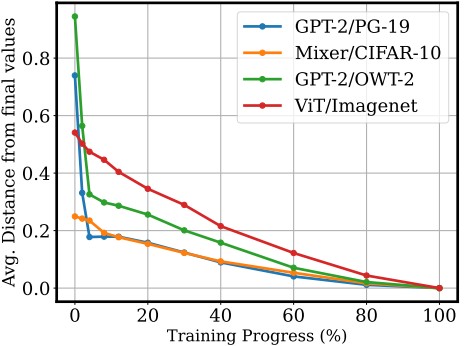

(a) Convergence distance $\Delta_t$ of residual weights $\boldsymbol{\alpha}$ during training.

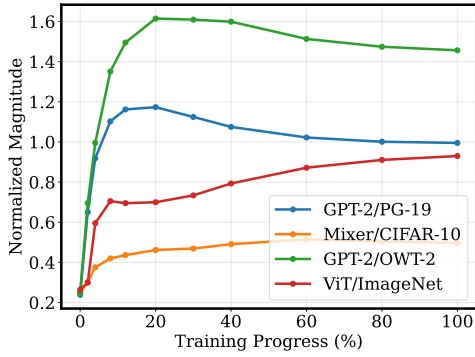

(b) Evolution of total connectivity strength $\Gamma$.

**Task difficulty.** To examine how task complexity affects connectivity patterns, we compute $\Gamma$ for networks trained on CIFAR-2, CIFAR-10, and ImageNet (1000 classes). Table 3 shows that $\Gamma$ increases with task difficulty, approaching the connectivity value of a vanilla residual network ($\Gamma = 1$) for the most complex tasks. However, unlike ResNets where connectivity is uniformly distributed (all $\alpha_i =$

| Dataset | Value |
|---|---|
| CIFAR-2 | 0.326 |
| CIFAR-10 | 0.428 |
| ImageNet | 0.930 |

Table 3: **Total connectivity strength** $\Gamma$ for increasingly complex image tasks.

---

[10]This experiment validates our choice of initialization values and suggests that architectural priors encoded in the initialization act as induction biases shaping final network connectivity.

[11]Appendix E visualizes the evolution of the connectivity matrix $\mathbf{C}$ also detailing the emergence of modularity shown in Fig. 1.

1), H-ACNs dynamically allocate connectivity strength based on learned residual weights. Adaptive allocation allows H-ACNs to match or exceed ResNet performance on complex tasks, while maintaining the flexibility to compress on simpler tasks.

**Connectivity patterns across modalities.** Our analysis reveals distinct connectivity patterns across modalities. Vision tasks (Figure 1) converge to connectivity matrices with stronger near-diagonal elements, suggesting preference for connections between adjacent layers. Language tasks develop connectivity matrices with distinct block structures, indicating selective long-range connections between specific layer groups. These domain-specific adaptations demonstrate that H-ACNs learn connectivity patterns tailored to different computational requirements, rather than converging to a universal architecture.

**Cognitive analogies.** These patterns exhibit intriguing parallels to brain organization: vision's local connectivity resembles the columnar organization of visual cortex (Felleman & Van Essen, 1991; Riesenhuber & Poggio, 1999), while language's modular blocks mirror the hierarchical structure of frontal-temporal language networks (Friederici, 2011). Moreover, the finding that task complexity increases connectivity strength aligns with brain studies showing that more demanding cognitive tasks recruit additional network connections and larger-scale integration across brain regions (Bassett et al., 2010). This suggests H-ACNs may discover connectivity principles that reflect domain-appropriate information processing, consistent with the hierarchical modularity observed in brain networks (Meunier et al., 2010).

**Computational analogies.** Our approach extends neural architecture search (NAS) by integrating auto-compressing inductive biases at initialization. We demonstrate that initializing H-ACNs closer to the ACN regime enables networks to learn connectivity patterns tailored to task complexity while maintaining training efficiency. This challenges the common practice of ResNet-like initialization of NAS and highlights the importance of architectural priors in shaping learned representations. Further comparisons with related work on learnable connectivity are provided in Appendix B.

## 5 CONCLUSION

We presented Hybrid Auto-Compressing Networks (H-ACNs), a unified architecture that interpolates between Auto-Compressing Networks and ResNets through trainable scalar residual weighting parameters, with ACNs and ResNets as special cases. H-ACNs achieve training efficiency comparable to ResNets while preserving the superior robustness, compression capabilities, and generalization of ACNs across vision transformers, MLP-mixers, and GPT-2 architectures. Learned residual weights exhibit modality- and task-specific connectivity patterns: vision tasks converge to local connectivity patterns, while language tasks develop modular hierarchical structures. Further, initialization near the ACN regime provides a crucial architectural prior that leads to better architectural choices. The emergence of domain-specific structures suggests that optimal architectural design should vary across modalities and tasks. This is a particularly promising direction for future work, namely, studying the functional connectivity patterns that emerge during training and working towards adaptive neural architecture design.

## 6 LIMITATIONS & BROADER IMPACT

Our evaluation focused on mid-scale experiments across a variety of tasks, models, and data modalities, providing initial evidence of the effectiveness and generality of our approach. However, scaling up the language modeling experiments to larger models and datasets is necessary to fully assess the robustness and applicability of our method in more demanding settings. Similarly, exploring modality-specific connectivity patterns in multimodal architectures could reveal additional insights into how adaptive connectivity can improve performance and efficiency across different types of data. While we explored various initialization schemes, the focus was primarily on auto-compressing versus residual initialization; a deeper study of more complex or structured initialization strategies remains as future work. Overall, our work aims to develop efficient and adaptive neural networks that adjust their computation and connectivity to the task, improving generalization and robustness. In Appendix F, we provide the Ethics and Reproducibility statement and elaborate on our use of LLM assistance.

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

## A  TRAINING DETAILS

**MLP-Mixer/CIFAR-10.**  We use a 16-layer architecture with a hidden size $d = 128$. The input image resolution is $32 \times 32$ with 3 channels, patches of size $4 \times 4$. The channel-mixing MLP dimension is set to $D_C = 512$, while the token-mixing dimension is $D_S = 64$. Training is performed with the AdamW optimizer Loshchilov & Hutter (2017), using a maximum learning rate of $0.001$, a cosine learning rate scheduler with warmup, and a batch size of 64.

**ViT/Imagenet.**  We used the setup described in (Beyer et al., 2022).

**GPT-2.**  We present in-detail our model and training hyperparameter choices in the table below:

| Model Hyperparam. | Value |
|---|---|
| Number of Heads | 12 |
| Number of Layers | 24 |
| Embedding Size | 768 |
| Vocab Size | 50304 |
| Sequence Length | 256 |
| Dropout | 0.2 |
| Positional Encoder | rotary |
| Tokenizer | GPT-2 |
| Number of Parameters | 208.54M |

Table 4: Models Configuration

| Training Hyperparam. | Value |
|---|---|
| Batch Size | 128 |
| Accumulation Steps | 4 |
| Iterations | 240,000 |
| Learning Rate | 0.001 |
| Warmup Percent | 0.05 |
| Weight Decay | 0.01 |
| $\beta_1$ | 0.9 |
| $\beta_2$ | 0.95 |
| Scheduler | cosine |
| Optimizer | AdamW |
| Gradient Clipping | 1.0 |
| Data Type | `torch.bfloat16` |
| Distributed Backend | NCCL |

Table 5: Training Configuration

## B  RELATED WORK

### B.1  MULTI-PATH ARCHITECTURES

The development of multi-path architectures has been a critical advancement in addressing optimization challenges in deep neural networks, especially vanishing and exploding gradients (Bengio et al., 1994). Highway Networks (Srivastava et al., 2015) introduced gated skip connections that enabled effective training of very deep models by facilitating signal flow. Residual Networks (ResNets) (He et al., 2016) simplified this design with identity skip connections, allowing deep models to be trained without introducing additional parameters. These architectures have been shown to improve gradient flow, smooth loss landscapes and enhance the gradient dynamics of deep networks (Zaeemzadeh et al., 2020; Li et al., 2018; Balduzzi et al., 2017). Furthermore, ResNets have been interpreted as implicit ensembles of shallower networks, offering multiple computational paths of varying depth (Veit et al., 2016). Building on the success of ResNets, a wide range of architectural variants have been proposed to increase representational capacity through richer feature fusion; DenseNets (Huang et al., 2017) replace addition-based fusion with concatenation to enable feature reuse across layers, while FractalNets (Larsson et al., 2016) use recursive structures to create deep ensembles.

More recently, research has shifted towards *learnable connectivity*, generalizing vanilla ResNets to architectures where inter-layer interactions are explicitly parameterized. Examples include learned weighted averaging across layer outputs (Pagliardini et al., 2024), attention-based inter-layer fusion (ElNokrashy et al., 2022), and hyper-connected modules (Zhu et al., 2024). These approaches can be formalized as:

$$input_k = \sum_{i=0}^{k-1} \mathbf{c}_{i \to k} \, h_i, \tag{6}$$

where $\mathbf{c}$ may be a learnable scalar, an input-dependent attention weight, or even a full matrix.

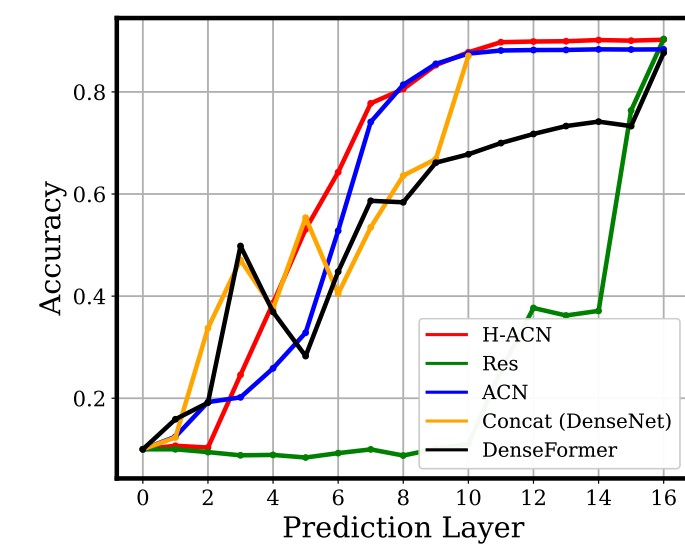

Figure 7: **MLP-Mixer/CIFAR-10.** We extend the experiment by including DenseFormer and DenseNet style Mixers.

In this work, we restrict $\mathbf{c}$ to scalar parameters, keeping computation comparable to ResNets and ACNs while focusing on how these weights are initialized. Unlike DenseFormers, which require $O(L^2)$ residual parameters and $O(L)$ extra memory to cache all intermediate outputs, we introduce only $L$ learnable residual weights. These act jointly to produce each $\mathbf{c}$, yielding a far more parameter- and memory-efficient design.

We show that leveraging the *auto-compressing inductive bias* at initialization enables efficient learning of robust representations, with connectivity adapted to the task and modality. For comparison, we extend our MLP-Mixer/CIFAR-10 experiments with a DenseFormer-style mixer and a DenseNet (using concatenations instead of additions)[12]. The results presented in Fig 7 show that H-ACNs, initialized close to ACNs, achieve performance on par with vanilla ResNets at the same training speed, while surpassing all other architectures. This is achieved without additional memory overhead and with only 16 extra parameters. Moreover, H-ACNs display strong auto-compression, akin to ACNs, revealing redundancy in the predefined architecture.

### B.2 PRUNING & DYNAMIC COMPUTATION

Another line of work includes *pruning-based* methods (Cheng et al., 2024; Frankle & Carbin, 2018; Sanh et al., 2020; Konstantaropoulos et al., 2025), which remove redundant weights or connections to achieve architectural compression, and *dynamic computation* methods (Han et al., 2021; Matsubara et al., 2022), which dynamically adjust computation based on the input. ACNs have already demonstrated that their auto-compression synergizes with these approaches, leading to stronger performance vs. inference-efficiency trade-offs compared to ResNets. H-ACNs follow the same principle: as we show in the main paper, intermediate layer performance is significantly better than ResNets and comparable to ACNs, suggesting that analogous advantages in efficiency and performance can be expected.

---

[12]Because concatenations increase parameters per layer, we use a 10-layer DenseNet with a parameter count comparable to the 16-layer counterparts.

## C Inter-Layer Connectivity Matrix of standard Architectures

Here, we show the **Direct Layer Connectivity Matrix**, as defined in the main paper, of FFNs, ResNets and ACNs. As explained, $C[i][j] = c_{i \to j}$ denotes the direct connection from source layer $i$ to target layer $j$. The matrices are shown in Fig. 8.

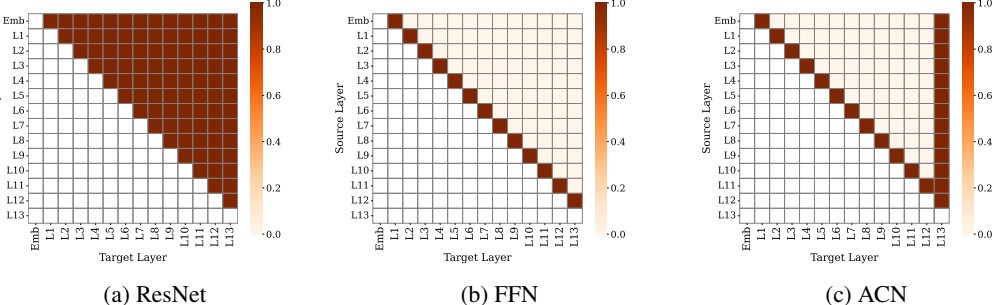

(a) ResNet  (b) FFN  (c) ACN

Figure 8: Connectivity patterns across architectures.

## D Ablation Studies

In this section, we further ablate the initialization of the $\alpha$ residual weights and provide a detailed explanation of the Depth-Adaptive LayerNorm (DepthLN) technique used in our experiments. Due to resource and time constraints, the ablation studies were performed on PG-19 with training on 6B tokens.

**Initialization of alphas.** First, we ablate the choice of initialization for the residual weights. We vary the mean of the normal distribution used for initialization, as defined in the main paper, and also test alternative strategies: (1) Half layers, using a normal with $\mu = 0.4$ for the first half of the layers and $\mu = 0.15$ for the rest; and (2) Exp. decay, where the mean decays exponentially from 0.25 to 0.1 with increasing depth. We present the results below:

| Initialization | PPL |
|---|---|
| $N(0.15, 0.01)$ | $19.73 \pm 0.15$ |
| $N(0.25, 0.01)$ | $19.25 \pm 0.03$ |
| $N(0.15, 0.005)$ | $19.64 \pm 0.04$ |
| $N(0.25, 0.005)$ | $\mathbf{19.22} \pm 0.02$ |
| Half layers: $N(0.4, 0.005)/N(0.15, 0.005)$ | $19.48 \pm 0.05$ |
| Exp. decay: $N(0.25, 0.005) \to N(0.1, 0.005)$ | $20.05 \pm 0.04$ |

Table 6: Ablation of different residual weight initializations.

Among all tested initializations, $N(0.25, 0.005)$ achieves the best performance, despite being uniform across layers. Initialization strategies based on depth did not provide any improvement. As noted in the main paper, a more detailed exploration of initialization strategies is left for future work.

**Depth-adaptive LayerNorm (DepthLN).** For a layer of dimension $d$ at depth $l$, let the standard LayerNorm of input $x_l \in \mathbb{R}^d$ be $\text{LN}(x_l)$. Then, Depth-adaptive LayerNorm scales the normalized output by a learnable depth-dependent scalar $\alpha_l$:

$$\text{DepthLN}(x_l) = \alpha_l \cdot \text{LN}(x_l), \quad \alpha_l = 1 + l \cdot s, \tag{7}$$

where $s$ is a small learnable strength factor initialized at $0.05$. This introduces only *one extra parameter per layer*, enhancing forward signal flow without significant computational overhead. Ablation experiments (Table 7) show that including DepthLN consistently improves H-ACN performance.

| Model | PPL |
|---|---|
| $N(0.15, 0.01)$ | $19.73 \pm 0.15$ |
| $\hookrightarrow$ w/o DepthLN | $20.5 \pm 0.2$ |
| $N(0.25, 0.01)$ | $19.25 \pm 0.03$ |
| $\hookrightarrow$ w/o DepthLN | $19.47 \pm 0.08$ |

Table 7: Ablation of Depth-adaptive LayerNorm.

## E    EVOLUTION OF THE DIRECT INTER-LAYER CONNECTIVITY MATRIX

In this section, we provide a detailed visualization of the evolution of the direct layer connectivity matrix $C$ during GPT-2 pre-training on PG-19. As shown in the figure below, we can observe (1) the sequential formation of modules during training, with the first two modules emerging initially, followed by the final one, and (2) that, consistent with the metrics reported in the main paper, the connectivity structure is largely established by 50–60% of training, at which point the modules are clearly defined.

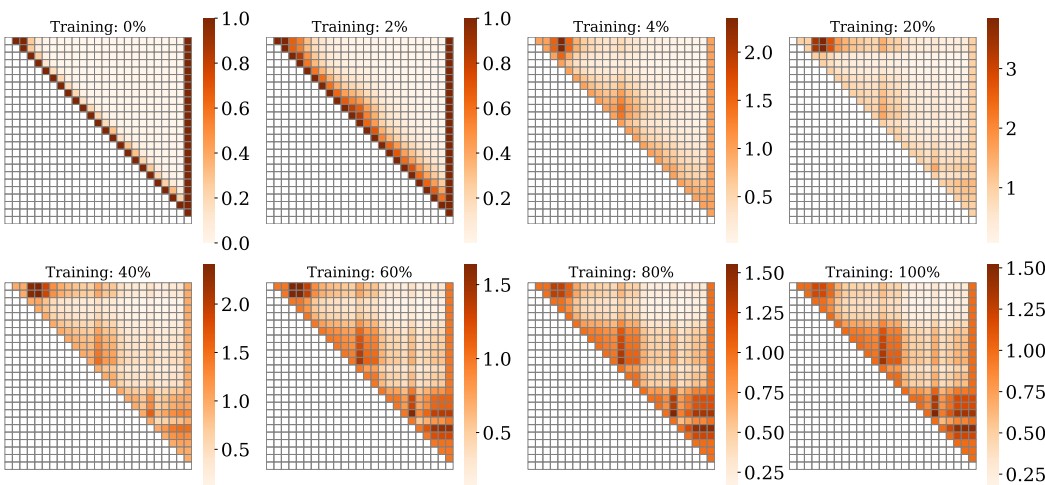

Figure 9: Evolution of the direct layer connectivity matrix during training of GPT-2 decoder on PG-19.

## F    ETHICS, REPRODUCIBILITY, AND LLM USAGE

**Ethics Statement.** The authors affirm that they have read and will adhere to the ICLR Code of Ethics in all aspects of this work.

**Reproducibility Statement.** All necessary implementation details for reproducibility are presented (model architectures, we use public datasets, all training details and hyperparameter choices are provided) and a detailed description of the techniques used in this work (in the main paper and Appendix). We also plan to make the code publicly available.

**Use of Large Language Models.** Large language models were employed to assist in polishing the manuscript and help with grammar. All content has been carefully reviewed and adjusted by the authors, who take full responsibility for the final published work.

# G  FORWARD PASS OF H-ACNS

Here, we present a pseudo-implementation of the forward pass of H-ACNs, to further show the interpolation between ACNs and ResNets:

---

**Algorithm 1** Forward pass of ACNs **(a=0)** and ResNets **(a=1)**

---

1: $x \leftarrow \text{emb}(input)$
2: $current \leftarrow x$
3: **for** each i, `layer` in **enum**(`layers`) **do**
4:     $x_{\text{out}} \leftarrow \text{layer}(x)$
5:     $current \leftarrow current + x_{\text{out}}$        # Long Connections to the output are fixed
6:     $x \leftarrow x_{\text{out}} + \mathbf{a}_i \cdot x$        # Short skip connections are weighted
7: **end for**
8: $x_{\text{cls}} \leftarrow current$
9: $x_{\text{cls}} \leftarrow \text{cls}(x_{\text{cls}})$

---

# H  INTER LAYER CONNECTIVITY VS LAYER FUSION

Most previous works on learnable residual architectures (like DenseFormer (Pagliardini et al., 2024), Depth-Wise Attention (ElNokrashy et al., 2022)) focus on **layer fusion**: how to weight or combine/fuse outputs from layers that are already fully connected. These methods preserve the classical ResNet assumption that every layer is directly connected to all subsequent layers, and the learnable parameters simply determine how these signals are fused (e.g., via weighted averaging, attention, or concatenation). Fusion mechanisms therefore operate *within* an all-to-all connectivity pattern. In contrast, following ACNs, we focus on **inter-layer connectivity**: determining *which layers should directly connect* in the first place. This represents a distinct and orthogonal architectural dimension compared to fusion and both angles are crucial for network behavior. In our case, as already shown in the as evidenced by the FFN to ResNet or ResNet to ACN transition, inter-layer connectivity structure fundamentally shapes information routing and gradient propagation.

**ResNet-like initialization prevents exploration of connectivity.** Importantly, existing learnable residual methods implicitly assume—and initialize close to—the classic ResNet regime, where all connections start equally active. Under such initializations, networks tend to remain ResNet-like throughout training. Belo, we present key evidence supporting this claim:

- Prior work, specifically in the DenseFormer paper Fig.5, shows that learned connectivity remains nearly all-to-all, with additional increased weight on the input.

- In our unified formulation, initializing scalar connectivity weights to the residual regime ($\alpha = 1$ for all layers) leads to trained weights that stay extremely close to 1, indicating that the model does not move away from the ResNet-like connectivity pattern. Specifically, the converged alphas are:

$$\begin{bmatrix} 0.99, 0.62, 1.00, 1.00, 0.95, 0.95, 0.98, 0.93, 0.94, 0.92, 0.94, 0.93 \\ 0.90, 0.90, 0.94, 0.95, 0.94, 0.95, 0.92, 0.94, 0.96, 1.00, 1.00, 1.00 \end{bmatrix}$$

- We further verified this by training DenseFormer under our setup and observed the same behavior.

These findings challenge current practice: the field has largely optimized fusion mechanisms, but has not explored the space of inter-layer connectivity itself. ResNet-like initialization strongly biases the model toward the all-to-all regime, preventing it from discovering alternative connectivity structures that may lead distinct learned representations (e.g. ACNs).

That said, we extend our experimental setup to systematically evaluate how different initialization strategies affect both the learned connectivity patterns and downstream performance under different

| Init Method | PPL |
|---|---|
| $\mathcal{O}(L)$ **Models** | |
| H-ACN ($\mu = 0.25$) | $19.79 \pm 0.015$ |
| Res | $19.84 \pm 0.01$ |
| H-ACN ($\mu = 0.5$) | $19.97 \pm 0.01$ |
| H-ACN ($\mu = 0.75$) | $20.00 \pm 0.01$ |
| H-ACN ($\mu = 1.0$) | $19.93 \pm 0.005$ |
| H-ACN w/o long ($\mu = 1.0$) | $19.76 \pm 0.01$ |
| H-ACN w/o long ($\mu = 0$) | $19.80 \pm 0.01$ |
| $\mathcal{O}(L^2)$ **Models** | |
| DenseFormer | $19.74 \pm 0.015$ |
| LayerComb (Res-init) | $19.63 \pm 0.03$ |
| LayerComb (ACN-init) | $19.39 \pm 0.02$ |

Table 8: Perplexity across initialization strategies for both linear- and quadratic-connectivity families.

layer-fusion mechanisms. The results in Table 6 [13] indicate that ACN-like initialization, as proposed in this work, can lead to improved performance compared to the standard ResNet-style initialization. Notably, an $\mathcal{O}(L^2)$ learnable architecture—similar to DenseFormer—achieves the best perplexity when initialized with our ACN-like scheme. This provides key evidence that layer fusion and inter-layer connectivity (i.e., which layers connect to which) are orthogonal architectural dimensions.

# I   DEPTH-ADAPTIVE TRANSFER LEARNING

A central claim of ACNs is their ability to dynamically allocate depth based on task difficulty: easier tasks naturally use fewer layers, emerging directly from the training dynamics. This leads to an important question: *Does this depth-adaptive behavior persist when ACNs are fused with ResNets under the H-ACN formulation?*

To examine this, we fine-tune our pretrained ImageNet ViT models on CIFAR-10. As shown in Fig. 10, H-ACNs continue to modulate their effective depth during downstream training. Specifically, the top three layers—important for ImageNet—become redundant for the simpler CIFAR-10 task and can be pruned with negligible impact on accuracy, reducing inference cost and latency.

This indicates that H-ACNs develop a more hierarchical organization of representations than standard residual architectures, enabling them to rely on fewer layers for simpler tasks. This opens a promising direction: large pretrained models may utilize full depth during large-scale pretraining, yet *naturally adapt* their depth to downstream tasks without requiring external pruning procedures. Furthermore, such adaptive compression can support improved early-exit behavior, enabling additional latency gains, as already shown in the original ACN paper.

---

[13] H-ACN w/o long is essentially the instantiation of various learnable residual works. LinearComb is a learnable architecture with $L^2$ parameters that combine the previous layer outputs **after the MLP** and not after the residual addition (full block) as in the case of the Denseformer.

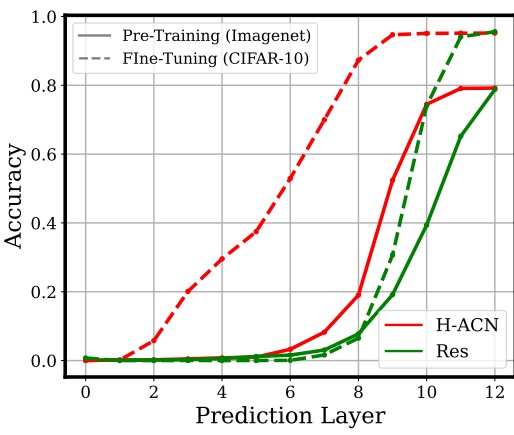

Figure 10: Depth allocation during fine-tuning from ImageNet to CIFAR-10. H-ACNs reduce reliance on upper layers for the simpler downstream task, enabling natural depth compression.

