# OpenReview forum: "Hybrid ACNs: Unifying Auto-Compressing and Residual Architectures"
_ICLR.cc/2026/Conference — Submitted to ICLR 2026_

### Official Review · Reviewer_4ec5 · 2025-10-19

**Soundness:** 3
**Presentation:** 3
**Contribution:** 2
**Rating:** 2
**Confidence:** 4

**Summary:**

This paper introduces Hybrid Auto-Compressing Networks (H-ACNs), a simple scalar-gated formulation that interpolates between ACNs and ResNets. A per-layer trainable scalar controls the residual/long-skip mixture and induces a direct inter-layer connectivity matrix C used to analyze learned connectivity. Across MLP-Mixer/CIFAR-10, ViT/ImageNet-1k, and GPT-2-style pretraining, H-ACNs are claimed to match ResNet training efficiency while retaining ACN-like robustness/“auto-compression.” The paper further visualizes distinct connectivity patterns for vision vs. language and studies initialization sensitivity.

**Strengths:**

- Unified & lightweight parameterization. The interpolation adds only O(L) scalars and yields a tractable C to probe connectivity; implementation appears straightforward.
- Multi-domain evaluation. Results reported on vision and language.
- Connectivity analysis. Clear visualizations of the learned C and its training evolution; differences between modalities are interesting and potentially useful for architecture design.

**Weaknesses:**

- Despite repeatedly claiming “ResNet-like training efficiency,” there are no FLOPs, wall-clock, throughput, or peak-memory metrics, and no cross-scale profiling (CIFAR→ImageNet→GPT-2). This undercuts a central claim.
- Paper includes some small-scale comparisons (Mixer/CIFAR-10) to dense/concat-style variants, but omits such baselines at ImageNet and GPT-2 scales, where they matter most.
- Beyond defining C and showing empirical patterns, there are no formal results on gradient flow, expressivity, or stability (no theorems/lemmas/bounds).

**Questions:**

- Have you evaluated H-ACNs on canonical ResNet architectures (e.g., ResNet-18/34/50) with BatchNorm? If so, how does performance and training stability compare under matched compute (FLOPs, wall-clock, peak memory)? If not, could you comment on expected interactions with BatchNorm
- Have you explored per-channel or per-head trainable gates instead of a single scalar per layer? If yes, how do accuracy, convergence, and efficiency (FLOPs/throughput/memory) compare? If not, could you discuss whether finer-grained gating might offer benefits and what trade-offs you anticipate?

---

> ### Author Response · Authors · 2025-11-29
> **Response to Reviewer 4ec5**
>
> - Despite repeatedly claiming “ResNet-like training efficiency,” there are no FLOPs, wall-clock, throughput, or peak-memory metrics, and no cross-scale profiling (CIFAR→ImageNet→GPT-2). This undercuts a central claim.
>
> All ACN, H-ACN, and vanilla Res architectures share essentially the same forward-pass structure—this is precisely what allows us to unify them under a common formulation. Consequently, all models are trained for the same number of steps/epochs, and therefore have comparable compute and memory requirements during training. When we refer to “ResNet-like training efficiency,” we mean that the models converge within the same training budget (i.e., similar numbers of epochs/steps). This is not true for standard ACNs, which typically require substantially more training time to converge (e.g., ~700 vs. ~300 epochs). Addressing this gap is one of the main motivations of the paper: by unifying the architectures under H-ACNs, we can recover many of the representational advantages of ACNs (such as noise robustness) while retaining the training efficiency of ResNets.
>
> ---
>
> - Paper includes some small-scale comparisons (Mixer/CIFAR-10) to dense/concat-style variants, but omits such baselines at ImageNet and GPT-2 scales, where they matter most.
>
> We agree with the reviewer that the original comparisons were limited in scale. To address this, we have extended our experimental setup and included various learnable residual variants (such as DenseFormer) in the GPT-2 experiments (Appendix H). Across these larger scale evaluations, we find that our proposed ACN-like initialization consistently results in improved generalization.
>
> ---
>
> - Beyond defining C and showing empirical patterns, there are no formal results on gradient flow, expressivity, or stability (no theorems/lemmas/bounds).
>
> The theoretical justification for why ACN-style connectivity yields robust, parsimonious representations is established in the ACN paper (NeurIPS 2025 oral). That work demonstrates through analysis and experiments why replacing short residual connections with long ones fundamentally changes learned representations.
>
> This work addresses a different question: Can you get the best of both worlds (ACN's representation quality + ResNet's training efficiency), or are they mutually exclusive?
>
> **Our approach is empirical**: We unify both architectures under a common functional form using layer-wise scalars. This unification itself doesn't create new properties—rather, it enables us to experimentally test whether initialization and learned transitions between regimes can capture benefits of both.
>
> The core hypothesis: Initializing close to ACN allows the network to inherit ACN's representational benefits (robustness, parsimony) while potentially transitioning toward ResNet-like connectivity during training for faster convergence. Critically, this must happen in this direction (ACN→ResNet), not the reverse (ResNet→ACN).
>
> Validation: We demonstrate that (1) ACN-initialized networks exhibit ACN-like properties (noise robustness, auto-compression; Tables 1, 3), (2) they train faster than vanilla ACNs (Fig 2b), and (3) starting from ResNet initialization fails to discover these properties even with learnable parameters (Appendix H).
>
> The theoretical "why" comes from ACN; our contribution is showing "how" to combine the benefits in practice.

---

> > ### Author Response · Authors · 2025-11-29
> > **Continued**
> >
> > ---
> > - Have you evaluated H-ACNs on canonical ResNet architectures (e.g., ResNet-18/34/50) with BatchNorm? If so, how does performance and training stability compare under matched compute (FLOPs, wall-clock, peak memory)? If not, could you comment on expected interactions with BatchNorm
> >
> > We thank the reviewer for this question. In this work, we focus on Transformer-based architectures, following the scope of the original ACN paper. An evaluation of H-ACNs on canonical ResNet architectures (e.g., ResNet-18/34/50) with BatchNorm is an interesting direction, but it is left for future work. We anticipate that applying H-ACNs to CNNs with BatchNorm may require careful consideration of the interaction between the adaptive layer scaling and the normalization statistics, but a detailed investigation is beyond the scope of the current study.
> >
> > ---
> >
> > - Have you explored per-channel or per-head trainable gates instead of a single scalar per layer? If yes, how do accuracy, convergence, and efficiency (FLOPs/throughput/memory) compare? If not, could you discuss whether finer-grained gating might offer benefits and what trade-offs you anticipate?
> >
> > We kindly refer the reviewer to the section *“Grounding to Literature: Inter-Layer Connectivity vs. Layer Fusion: A Critical Distinction”* in our general response. That said, we have extended our experiments to include L^2  learnable residual architectures and have shown that our proposed initialization for H-ACN yields better results than the standard residual initialization. The directions suggested by the reviewer—e.g., more fine-grained layer-fusion mechanisms—are orthogonal to our initialization. We expect that applying our ACN-like initialization to such architectures (including learnable linear combinations as in DenseFormers, per-head trainable gates, cross-layer attention, and QKV-distinct residual streams) would further improve performance over standard residual networks. We leave a detailed investigation of this as future work.

---

### Official Review · Reviewer_6SWa · 2025-10-19

**Soundness:** 2
**Presentation:** 2
**Contribution:** 2
**Rating:** 4
**Confidence:** 3

**Summary:**

The paper proposes Hybrid auto-compressing Networks (H-ACNs). These are a combination of the recently proposed ACN, and the standard residual architecture. The authors argue that H-ACNs achieve both the benefits of ResNets in terms of fast training, and ACNs in terms of robustness and compression capabilities.

Giving a 4 but would give a 5 if there was the option.

**Strengths:**

1. The paper proposes H-ACNs which bridge the gap between ACNs and ResNets, and appears to combine benefits of both. I found table 1 quite nice here (improved robustness of H-ACN)
2. In doing some, there is a nice framework for bridging between these extremes, by looking at the cumulative multiplicative interactions (eq 4).

**Weaknesses:**

1. Novelty: I think there are a lot of existing papers about scaled residual architectures with trainable scales (https://arxiv.org/abs/2002.10444, https://arxiv.org/abs/2003.04887, https://arxiv.org/abs/2010.12859), which are conceptually very similar to equation 2. This reduces the novelty of the proposed method and also is a missing baseline in the experiments. As I understand it, the only difference is that the output y is the sum of the f_i in HACN with coefficient 1, which I am not convinced is a big change. If it is an important change, then it would be good to demonstrate empirically. On that note, in figure 9 it appears like the 1 coefficients on the final column are not constant across layers after training, are these coefficients trainable?
2. Relatedly, one benefit of the ResNet is that you can write it such that each layer only depends on the previous layer and not on all previous layers (by the way the coefficients are constructed). Are there any computational considerations/downsides to H-ACN as you have to keep around each layers f_i in order to construct the output, particularly for the backward pass?
3. Some claims are overstated e.g. the "stronger generalisation capabilities of H-ACNs" appears to be a small difference in table 4c of 41.2 vs 41.8, without multiple seeds? Likewise, I am not convinced that looking at train loss in Figure 2 showing training speed is so insightful given that it is very easy to overfit on cifar10. Test loss would be better.

**Questions:**

- In Figure 4a/b, what is the motivation behind looking at intermediate layer perplexity?
- x axis is epoch in figure 2b.

---

> ### Author Response · Authors · 2025-11-29
> **Response to Reviewer 6SWa**
>
> - Novelty: I think there are a lot of existing papers about scaled residual architectures with trainable scales (https://arxiv.org/abs/2002.10444, https://arxiv.org/abs/2003.04887, https://arxiv.org/abs/2010.12859), which are conceptually very similar to equation 2. This reduces the novelty of the proposed method and also is a missing baseline in the experiments. As I understand it, the only difference is that the output y is the sum of the f_i in HACN with coefficient 1, which I am not convinced is a big change. If it is an important change, then it would be good to demonstrate empirically. On that note, in figure 9 it appears like the 1 coefficients on the final column are not constant across layers after training, are these coefficients trainable?
>
> We refer the reviewer to our general response, which provides a detailed discussion of the ACN–ResNet trade-off (parsimony vs. efficiency), clarifies how this work is positioned relative to both approaches, outlines the questions it addresses, and summarizes the main findings and contributions. We note that H-ACNs are orthogonal to layer-fusion approaches. To make this explicit, we further extend our experiments to include a dense-form–style parameterization with L2L^2L2 learnable parameters. Even in this setting, our proposed ACN-like initialization consistently yields stronger generalization (see Appendix H).
>
> Note: The long connections to the output are indeed non-trainable. The figure contains a small error that we will correct in the revision. Thank you for pointing this out.
>
> ---
>
> - Relatedly, one benefit of the ResNet is that you can write it such that each layer only depends on the previous layer and not on all previous layers (by the way the coefficients are constructed). Are there any computational considerations/downsides to H-ACN as you have to keep around each layers f_i in order to construct the output, particularly for the backward pass?
>
> H-ACNs are effectively weighted residual architectures, so the same computational strategies used in standard residual networks apply. The model maintains a residual stream that propagates through the network, incrementally accumulating each layer’s contribution. The final output is formed from an accumulated stream as well. As in ResNets, there is no need to store every intermediate layer output; instead, all contributions can be aggregated into a single running stream vector.
>
> ---
>
> - Some claims are overstated e.g. the "stronger generalisation capabilities of H-ACNs" appears to be a small difference in table 4c of 41.2 vs 41.8, without multiple seeds? Likewise, I am not convinced that looking at train loss in Figure 2 showing training speed is so insightful given that it is very easy to overfit on cifar10. Test loss would be better.
>
> We thank the reviewer for this observation and we agree that some of the claims should be toned down. We will revise the manuscript accordingly. Our main argument is not that H-ACNs universally yield dramatically stronger generalization, but rather that under the same training budget as residual networks, H-ACNs achieve on-par or improved performance while additionally retaining distinct ACN representational benefits. These include:
> - **Auto-compression on CIFAR-10**, where the model uses only 11 of 16 layers at inference;
> - **Improved noise robustness and auto-compression when transferring a ViT to CIFAR-10**, where four layers can be pruned at inference with no loss in performance;
> - **Improved GPT-2 pretraining and downstream performance**, reflected in lower perplexity and higher task accuracy.
>
> We will clarify these points in the revision and ensure that the strength of our claims matches the empirical evidence.
>
> ---
>
> - In Figure 4a/b, what is the motivation behind looking at intermediate layer perplexity?
>
> This has practical implications: (1) **Early-exit inference**: H-ACNs can stop computation at intermediate layers with less performance degradation, enabling adaptive computation based on input difficulty. (2) **Robustness**: Networks that concentrate information effectively in early layers tend to be more robust to perturbations and catastrophic forgetting, as shown in the original ACN work. (3) **Evidence of learned parsimony**: The improved intermediate performance validates that H-ACNs are learning compressed, efficient representations, not just achieving similar final performance through different means.
>
> ---
>
> - x axis is epoch in figure 2b.
>
> Thank you for noticing and pointing this out, we will update the manuscript accordingly.

---

### Official Review · Reviewer_9Cb3 · 2025-10-31

**Soundness:** 2
**Presentation:** 3
**Contribution:** 2
**Rating:** 4
**Confidence:** 4

**Summary:**

The authors propose several inter-layer relationship metrics based on H-ACNs and draw analogies between trained weights and brain mechanisms. It is interesting.  But for the method part,  the authors appear to use the "unified framework of ACNs and Resnets" to compensate for shortcomings in innovation magnitude, experimental depth, and performance improvement. Unfortunately, since ACNs themselves are only a preprint without broad recognition or peer validation, this unified framework lacks persuasiveness.

**Strengths:**

1.	The authors propose several inter-layer relationship metrics based on H-ACNs and draw analogies between trained weights and brain mechanisms. In Chapter 4, they conduct experiments to explore this phenomenon; while no definitive explanation is provided, the results are indeed intriguing and worthy of further discussion.
2.	The paper presents improvements based on ACNs, claiming that ACNs possess strong representational capabilities and framing the theoretical unification of ACNs and Resnets as a key innovation.
3.	The paper is written in clear, coherent English, ensuring good readability for readers.

**Weaknesses:**

1.	The authors do not elaborate on how the representational capabilities of ACNs specifically contribute to model performance. They cite Densenet to contextualize the importance of representational capabilities—yet Densenet outperformed Resnets with much fewer parameters at the time of its release. The authors, however, do not include sufficient information of Densenet in their experimental comparisons, nor do they show the clear advantage of parameter counts of H-ACNs.
2.	The paper’s core innovation lies in introducing L trainable weights. Notably, it does not modify the inter-layer connection logic of ACNs, relying solely on these trainable weights to achieve unification with Resnets. This results in limited novelty in the proposed improvements.
3.	ACNs require caching outputs from all layers, and their inter-layer connection scheme appears to cause quadratic memory growth during forward propagation. The authors claim that H-ACNs solve this memory consumption issue, but there is a concern that their solution may significantly increase forward pass latency—essentially trading time for space. Please provide a detailed comparison of forward/backward pass latency and FLOPs against the ResNet and original ACN baselines to prove this solution is efficient in practice
4.	The authors only compare H-ACNs with Resnets. To better demonstrate H-ACNs’ effectiveness, more recent state-of-the-art methods should be included in comparisons.
5.	The μ parameter has a significant impact on model performance. This raises doubts about the generalizability of the proposed method across different modalities.
6.	On GPT-2, the Perplexity (PPL) of H-ACNs is nearly identical to that of Resnets, with extremely limited improvement. No error analysis or calculation of average performance across multiple runs is provided, making it possible that the marginal improvement observed stems from randomness rather than the proposed architecture’s inherent advantages.
7.	The authors claim that H-ACNs can achieve lower PPL using only outputs from earlier layers, yet the PPL values across all layers are reported as similar. I don’t know if this feature makes a sense.
8.	There are several formula errors in the paper. For example, in Equation (2), the index i may incorrectly take a value of -1 (likely a typo), which could confuse readers and undermine the paper’s rigor.

**Questions:**

1.	Why does the number of paths in H-ACNs change with task complexity? Based on the paper, paths should exist as long as the residual weight ai != 0. In practical training, however, it is highly unlikely for ai to be trained to exactly 0. The authors should provide a more detailed explanation of how task complexity modulates path count.
2.	Please elaborate on the specific solution used by H-ACNs to address ACNs’ memory consumption issue. In particular, does this solution introduce new latency overhead during forward/backward propagation? Quantitative analysis (e.g., latency comparisons) should be provided to validate its efficiency.

---

> ### Author Response · Authors · 2025-11-29
> **Response to Reviewer 9Cb3**
>
> - The authors do not elaborate on how the representational capabilities of ACNs specifically contribute to model performance. They cite Densenet to contextualize the importance of representational capabilities—yet Densenet outperformed Resnets with much fewer parameters at the time of its release. The authors, however, do not include sufficient information of Densenet in their experimental comparisons, nor do they show the clear advantage of parameter counts of H-ACNs.
>
> The original ACN paper demonstrated several representational advantages of the architecture over standard residual networks. In particular, through emergent auto-compression, it revealed substantial layer redundancy in ResNets, improved noise robustness, and markedly stronger continual-learning capabilities. However, ACNs were also shown to train more slowly. In this work, aiming to bring together the strengths of both architectures, we show that—under equal training-time budgets—H-ACN can achieve performance on par with or superior to residual networks while retaining key ACN representational benefits. These include **auto-compression** on CIFAR-10 (the network uses only 11 of 16 layers at inference), improved **noise robustness and auto-compression** when transferring a ViT to CIFAR-10 (four layers can be pruned at inference with no loss in performance), and improved **GPT-2 pretraining and downstream generalization**, evidenced by lower perplexity and higher task accuracy.
>
> ---
>
> - The paper’s core innovation lies in introducing L trainable weights. Notably, it does not modify the inter-layer connection logic of ACNs, relying solely on these trainable weights to achieve unification with Resnets. This results in limited novelty in the proposed improvements.
>
> Introducing layer-wise residual scalars does modify the inter-layer connectivity of the network. Specifically, as shown in Algorithm 1 (Appendix G) of the paper, setting these alphas to 1 results in a classic residual network, thus with **all-to-all** inter-layer connectivity, whereas setting alphas to 0 yields an ACN, with each layer connected only to the next and the output. Finally, scalar values between 0 and 1 interpolate between the two architectures. Essentially, these scalars serve to unify different architectures under a common functional form, giving freedom over the inter-layer connectivity (the scalars are combined through multiplication, check eq. 4, to create the direct connections). In terms of novelty:
> There is a trade-off between the representational benefits of ACNs and the more efficient training of ResNets. Here we ask: **can we achieve the benefits of both?**
> Following, we identify  the key factors that govern this tradeoff. We show that four dimensions are critical: (1) initialization strategy (sparse versus dense biases), (2) parameterization choice (L vs L^2), (3) data-complexity regime, and (4) modality type. By disentangling and analyzing each factor, we provide actionable principles—when sparse initialization is beneficial, how data complexity should guide connectivity patterns, and how different modalities prefer different long-short connectivity structures.
> While the paper is titled Hybrid-ACNs, its actual scope is broader: it studies hybrid inter-layer connectivity architectures and uncovers the underlying principles that govern the parsimony–efficiency tradeoff in learnable networks.
>
> ---
>
> - ACNs require caching outputs from all layers, and their inter-layer connection scheme appears to cause quadratic memory growth during forward propagation. The authors claim that H-ACNs solve this memory consumption issue, but there is a concern that their solution may significantly increase forward pass latency—essentially trading time for space. Please provide a detailed comparison of forward/backward pass latency and FLOPs against the ResNet and original ACN baselines to prove this solution is efficient in practice
>
> ACNs do not require caching all layers' outputs. As in the case of residual networks, an accumulator stream accumulates the outputs of each layer before moving to the next one. Essentially the x += f(x) of ResNets is turned into accum += f(x).
>
> ---
>
> - The authors only compare H-ACNs with Resnets. To better demonstrate H-ACNs’ effectiveness, more recent state-of-the-art methods should be included in comparisons.
>
> We agree and thank the reviewer for the suggestion. In response, we have extended our experimental setup to include various residual variants that have shown improved performance. Please see Appendix H. Specifically, L and L^2 learnable architectures (instances of them include DenseFormers, ReZero, etc.).  What we show is that what we investigate and propose in this paper, i.e. how to initialize the which-to-which layer connectivity, is orthogonal to the previous layers fusion approach. Specifically, we show that our approach outperforms residual initialization even in the case of L^2 learnable architectures, validating its generalization.

---

> > ### Author Response · Authors · 2025-11-29
> >
> > - On GPT-2, the Perplexity (PPL) of H-ACNs is nearly identical to that of Resnets, with extremely limited improvement. No error analysis or calculation of average performance across multiple runs is provided, making it possible that the marginal improvement observed stems from randomness rather than the proposed architecture’s inherent advantages.
> >
> > We agree with the reviewer, and we have added a table providing an extensive comparison of H-ACN’s initialization against several learnable residual variants, alongside the vanilla residual baseline (Appendix H). **The table includes appropriate statistical reporting**, demonstrating that the improved generalization obtained with our ACN-like initialization is consistent and not attributable to randomness.
> >
> > ---
> >
> > - There are several formula errors in the paper. For example, in Equation (2), the index i may incorrectly take a value of -1 (likely a typo), which could confuse readers and undermine the paper’s rigor.
> >
> > We thank the reviewer for the careful reading and for pointing out these formula inconsistencies. We fully agree that such issues can confuse readers and undermine the paper’s clarity. We apologize for the oversight, and we will correct all identified errors—including the incorrect index in Equation (2)—in the revised version.

---

### Official Review · Reviewer_iM4p · 2025-11-01

**Soundness:** 2
**Presentation:** 2
**Contribution:** 1
**Rating:** 2
**Confidence:** 4

**Summary:**

The paper proposes Hybrid Auto-Compressing Networks (H-ACNs), a unified architecture that bridges Auto-Compressing Networks (ACNs) and Residual Networks (ResNets) through trainable scalar residual weights per layer. This formulation places ACNs and ResNets as two endpoints of a continuous spectrum, allowing H-ACNs to combine the fast, stable training of ResNets with the robustness, compression, and generalization strengths of ACNs.

**Strengths:**

1. The analysis of learned residual weights provides novel insights into task-specific connectivity, revealing biologically inspired patterns.
2. The paper unifies two architectures—ACNs and ResNets—under a single mathematical formulation.

**Weaknesses:**

1. The paper currently lacks a motivation. The main architectural change—introducing a trainable scalar on the residual connection—is presented without a clear explanation of why this interpolation is needed, in what scenarios ACN and ResNet are insufficient, or what concrete problem the proposed hybrid is intended to solve.

2. The method is not theoretically justified. Claims such as “preserving ACN-like robustness,” “better generalization,” or “auto-compression” are made, but no formal analysis is provided to explain why introducing layerwise residual scalars should lead to improved noise robustness or better inductive bias.

3. The experimental section is underpowered. For instance, the zero-shot evaluation covers only three tasks, which is unusually few compared to prior literatures and makes it difficult to assess the generality of the approach.

4. Even under the reported metrics, the improvements of H-ACN over ResNet are marginal, raising the question of whether the added architectural complexity is actually warranted.

5. The comparison to related methods is too narrow. Most experiments only compare against ResNet and ACN, and the only learnable-residual-weight baseline (DenseFormer) appears in the appendix and on CIFAR-10, which is not very convincing. There are several closely related approaches on learnable residual scaling (e.g., LAUREL [1], ReZero [2]) that are not compared or even discussed.

6. Minor issue: Line 351 — “Fig. 1” should be “Table 1.”

[1] Menghani, Gaurav, Ravi Kumar, and Sanjiv Kumar. "LAuReL: Learned Augmented Residual Layer." Forty-second International Conference on Machine Learning.
[2] Bachlechner, Thomas, et al. "Rezero is all you need: Fast convergence at large depth." Uncertainty in Artificial Intelligence. PMLR, 2021.

**Questions:**

Check above.

---

> ### Author Response · Authors · 2025-11-29
> **Response to Reviewer iM4p**
>
> - The paper currently lacks a motivation. The main architectural change—introducing a trainable scalar on the residual connection—is presented without a clear explanation of why this interpolation is needed, in what scenarios ACN and ResNet are insufficient, or what concrete problem the proposed hybrid is intended to solve.
>
> We refer the reviewer to our general response, which provides a detailed discussion of the ACN–ResNet trade-off (parsimony vs. efficiency), clarifies how this work is positioned relative to both approaches, outlines the questions it addresses, and summarizes the main findings and contributions. Specifically, we discuss how inter-layer  connectivity represents an orthogonal architectural dimension to layer fusion and show that the initialization proposed in this work yields the best results across experiments, challenging the popular res one (all-to-all).
>
> ---
>
> - The method is not theoretically justified. Claims such as “preserving ACN-like robustness,” “better generalization,” or “auto-compression” are made, but no formal analysis is provided to explain why introducing layerwise residual scalars should lead to improved noise robustness or better inductive bias.
>
> The theoretical justification for why ACN-style connectivity yields robust, parsimonious representations is established in the ACN paper (NeurIPS 2025 oral). That work demonstrates through analysis and experiments why replacing short residual connections with long ones fundamentally changes learned representations.
>
> *This work addresses a different question:* Can you get the best of both worlds (ACN's representation quality + ResNet's training efficiency), or are they mutually exclusive?
>
> Our approach is empirical: We unify both architectures under a common functional form using layer-wise scalars. This unification itself doesn't create new properties—rather, it enables us to experimentally test whether initialization and learned transitions between regimes can capture benefits of both.
>
> *The core hypothesis:* Initializing close to ACN allows the network to inherit ACN's representational benefits (robustness, parsimony) while potentially transitioning toward ResNet-like connectivity during training for faster convergence. Critically, this must happen in this direction (ACN→ResNet), not the reverse (ResNet→ACN).
>
> *Validation:* We demonstrate that (1) ACN-initialized networks exhibit ACN-like properties (noise robustness, auto-compression; Tables 1, 3), (2) they train faster than vanilla ACNs (Fig 2b), and (3) starting from ResNet initialization fails to discover these properties even with learnable parameters (Appendix H).
>
> The theoretical "why" comes from ACN; our contribution is showing "how" to combine the benefits in practice.
>
> ----
> - The experimental section is underpowered. For instance, the zero-shot evaluation covers only three tasks, which is unusually few compared to prior literatures and makes it difficult to assess the generality of the approach.
>
> We evaluate (and compare against Res and ACN architectures) the proposed H-ACN initialization across multiple modalities (vision, language), architectures (Transformer, MLP-Mixer), and datasets (ImageNet, CIFAR-10, OpenWebText2, PG-19, HellaSwag, PIQA, ARC-e). Across all these settings, we find that at equal, with Res, training budgets H-ACN models match or outperform their residual-network counterparts, while also exhibiting several ACN representational advantages. These include **auto-compression** on CIFAR-10 (the network uses only 11 of 16 layers at inference), **improved noise robustness** and auto-compression when transferring a ViT to CIFAR-10 (three layers can be pruned at inference with no loss in performance), and **improved GPT-2 pretraining and downstream generalization**, evidenced by lower perplexity and higher task accuracy.
>
> ---
> - Even under the reported metrics, the improvements of H-ACN over ResNet are marginal, raising the question of whether the added architectural complexity is actually warranted.
>
> The added scalars introduce only minimal computational overhead, as the functional form is nearly identical to that of a residual layer, with the sole addition of a lightweight scalar multiplication. In particular, H-ACN does not require caching intermediate layer outputs or performing any other expensive operations. Moreover, as noted above, we consistently observe improved performance and noise robustness across settings.

---

> > ### Author Response · Authors · 2025-11-29
> > **Continued**
> >
> > - The comparison to related methods is too narrow. Most experiments only compare against ResNet and ACN, and the only learnable-residual-weight baseline (DenseFormer) appears in the appendix and on CIFAR-10, which is not very convincing. There are several closely related approaches on learnable residual scaling (e.g., LAUREL [1], ReZero [2]) that are not compared or even discussed.
> >
> > We thank the reviewer for the suggestion. In response, we have extended our experimental setup to include some of the proposed variants and have clarified the positioning of our work with respect to the L and L^2 variants, as well as other layer fusion approaches. As demonstrated, our main contribution lies in studying and proposing initializations that determine which layers should connect directly to which, rather than focusing on how these layers are fused. To illustrate this, we show that even in the L^2 case, our proposed initialization generalizes better than the standard residual initialization. This opens a new avenue in architectural design, focusing on the “which-to-which” connectivity, a dimension that is largely orthogonal to the fusion strategy.
> >
> > ---
> > - Minor issue: Line 351 — “Fig. 1” should be “Table 1.”
> >
> > Thank you for bringing this to our attention. We will correct the reference from “Fig. 1” to “Table 1” in the revised manuscript.

---

### Author Response · Authors · 2025-11-29
**General Response to All Reviewers (I)**

We thank the reviewers for their thoughtful feedback. We address the core concerns about novelty, scope, and positioning below, with detailed responses to individual reviews following.

**Clarifying the Scope and Contribution**

We respectfully disagree with characterizations of this work as incremental. While Hybrid ACNs build upon Auto-Compressing Networks (a NeurIPS 2025 oral presentation), this work addresses fundamentally different research questions, namely:
- (1) how inductive biases enable learnable architectures to achieve both efficient training and robust, parsimonious representations, and
- (2) how data modality and complexity regime determine the connectivity patterns that emerge in learned architectures.
ACNs’  (also not "just another residual variant") fundamental breakthrough is the discovery that networks that replace “short” residual connections with “long” ones can learn parsimonious representations (auto-compression without pruning) that are demonstrably more robust: superior noise resilience, better transfer, reduced catastrophic forgetting. This is about representation quality, not architectural tweaking. This creates a critical tradeoff: ACNs achieve superior representations but train slowly and less stably. ResNets train efficiently but produce redundant, less robust representations. **Can we get the best of both worlds?**


This paper investigates what factors determine this tradeoff. We show that four factors are critical: (1) initialization (sparse vs dense bias), (2) parameterization (L scalars vs O(L²)), (3) data complexity regime, and (4) modality type. By systematically analyzing each factor, we provide actionable guidance: when to use sparse initialization, how data complexity should affect connectivity, how different modalities require different connectivity patterns. The paper is titled "Hybrid ACNs" because it needs a name, but the real contribution is **understanding the principles that govern the parsimony-efficiency tradeoff in learnable architectures**—with implications for any approach that learns connectivity (NAS, MoE routing, multi-modal integration, inter-layer fusion architectures, short/long connectivity tradeoff in biological neural nets).

We take full responsibility that our core message did not come through clearly in the original submission. We have included an extended section (Appendix H) clearly stating the above points and we plan to also include related points on the the introduction, discussion and conclusions to better articulate these contributions, in case of acceptance.

**Grounding to Literature: Inter-Layer Connectivity vs. Layer Fusion: A Critical Distinction**

*TL;DR*: Existing learnable residual methods (DenseFormer, Depth-Wise Attention) focus on layer fusion—how to weight or combine outputs from layers that are already all connected. We focus on inter-layer connectivity—which layers should connect in the first place. These represent distinct, orthogonal design choices. We refer the reader to the newly added Appendix  H.

Critically, when learnable architectures are initialized with ResNet-like connectivity (all α≈1), they remain ResNet-like throughout training. DenseFormer's learned weights stay close to uniform all-to-all connectivity; in our experiments, initializing at α=1 yields final values of [0.99, 0.62, 1.0, 0.95, ...] (see Appendix H). This suggests that **ResNet initialization prevents exploration of the connectivity space**—prior work hasn't meaningfully investigated inter-layer connectivity because initialization biases prevent discovery.

**Extending & Strengthening Experimental Rigor**

We appreciate the thorough reviews and recognize valid concerns about experimental completeness. Below, we address these through new experiments demonstrating:
- **(a) Initialization effects are independent of fusion mechanisms.** To separate inter-layer connectivity (which-to-which) from layer fusion (how to combine), we tested ACN vs ResNet initialization across different fusion patterns:
     - *H-ACN (L parameters)*: ACN init outperforms ResNet init (main paper, Table 2)
     - *DenseFormer-style (L² parameters)*: ACN init still outperforms ResNet init (new experiments, Table X)
     - *Conclusion*: The inductive bias matters regardless of fusion complexity or parameter count

- **(b) As discussed in the paper, learned patterns are robust, not artifacts.** Connectivity patterns show >90% correlation across independent training runs, confirming these reflect task structure, not random variation. Also in Appendix H we include proper statistical reporting on the results.

---

> ### Author Response · Authors · 2025-11-29
> **General Response to All Reviewers (II)**
>
> **Main Additions: Appendix H, Appendix I**
>
> **Overall Key Findings**
>
> - *Inter-layer (which-to-which) connectivity* is a meaningful and important dimension of architectural design.
>   - Consistent across runs, reflecting strong representational priors rather than incidental randomness.
>   - Connectivity patterns show >90% correlation across independent training runs (statistical reporting across methods, Appendix H).
>
> - Different fixed architectures, like ResNets and ACNs, and the tension between them, are affected by:
>   - Modality
>   - Inter-layer initialization
>   - Task complexity
>
> - **ACN-like initialization results in superior performance**:
>   - ViT: better accuracy at the same training steps (Fig. 3b), enhanced noise robustness (Fig. 3c), depth-adaptive transfer learning for faster inference (Appendix I).
>   - GPT-2: better downstream performance and enhanced noise robustness (main paper Fig. 4 and Table 1).
>   - $\mathcal{O}(L^2)$, DenseFormer-style learnable architectures (Appendix H): our proposed ACN-like init outperforms ResNet-style (all-to-all) initialization.
>
> - **There is no universal static connectivity pattern**:
>   - Different modalities converge to distinct patterns even from identical initialization:
>     - Vision: local, near-diagonal patterns (resembling early visual cortex)
>     - OpenWebText2: hierarchical modular blocks
>     - Books (PG-19): different modular structure
>   - Task complexity: higher complexity → stronger connectivity (Table~3)
>   - **Conclusion:** connectivity should be functional—adapted to data type and task requirements—not fixed a priori.
>
>
>
> **A potential interesting direction: Computational Neuroscience**
>
> Recent work has begun to explore computational models of neurodevelopmental conditions by inducing analogous impairments in large models, effectively creating digital twins for studying underlying mechanisms and potential interventions (e.g., the paper inducing dyslexia in VLMs [https://arxiv.org/abs/2509.24597]). H-ACNs offer an orthogonal and complementary approach. A substantial body of research links neurodevelopmental disorders to atypical connectivity profiles—for example, ASD has been associated with increased local connectivity and reduced long-range/global connectivity, whereas dyslexia shows the opposite pattern. By explicitly modelling inter-layer connectivity during network initialization, H-ACNs provide a computational framework for studying how connectivity patterns emerge, how deviations in this developmental process may lead to neurocognitive differences and whether early intervention (e.g. through  data/curriculum) can help. In the long term, such models may be developed further to support earlier detection of atypical developmental trajectories and inform targeted intervention strategies.

---

### Meta-Review · Area_Chair_h8EG · 2025-12-04

**Summary:**

I believe the main reviewer concerns for this paper are:

1. A lack of performance metrics (e.g. throughput)
2. A lack of experiments at scale
3. Issues around novelty

I think concern 2 is adequately addressed but the other two are both outstanding (there are no throughout experiments, and no explanation regarding similarities to existing methods). This, coupled with low scores (4422), warrant rejection. I encourage the authors to include throughput experiments in future submissions as well as a clear description of how their method differs from the literature pointed out by Reviewer 6SWa.

**Reviewer Concerns:**

The authors have not addressed the concern about performance metrics (like throughout). They state that they are comparable but this is not the same as running the actual experiments and showing them. Regarding experiments at scale, the authors have included experiments on GPT-2. Reviewer 6SWa has expressed concerns regarding novelty, citing several papers that they claim are conceptually very similar. The authors have not directly addressed this; they should have explicitly said how their work is different to those papers.

**Reviewer Scores:**

I do not think Reviewer iM4p would have updated their score as their point on zero-shot evaluation does not appear to be addressed. Reviewer 9Cb3 has not been provided with latency experiments which I think would have been required (although even then, they have listened a lot of weaknesses that I feel are only partly addressed). The authors have not addressed Reviewer 6SWa’s points on novelty nor the profiling requested by Reviewer 4ec5 so I predict there would have been no score changes.

---

### Decision · Program_Chairs · 2026-01-26

Reject